

# 5-Methylindole kills various bacterial pathogens and potentiates aminoglycoside against methicillin-resistant *Staphylococcus aureus*

Zhongyan Li*, Fengqi Sun*, Xinmiao Fu and Yajuan Chen

Fujian Normal University, Fuzhou, Fujian, China
* These authors contributed equally to this work.

## ABSTRACT

Antibiotic resistance of bacterial pathogens has become a severe threat to human health. To counteract antibiotic resistance, it is of significance to discover new antibiotics and also improve the efficacy of existing antibiotics. Here we show that 5-methylindole, a derivative of the interspecies signaling molecule indole, is able to directly kill various Gram-positive pathogens (*e.g.*, *Staphylococcus aureus* and *Enterococcus faecalis*) and also Gram-negative ones (*e.g.*, *Escherichia coli* and *Pseudomonas aeruginosa*), with 2-methylindole being less potent. Particularly, 5-methylindole can kill methicillin-resistant *S. aureus*, multidrug-resistant *Klebsiella pneumoniae*, *Mycobacterium tuberculosis*, and antibiotic-tolerant *S. aureus* persisters. Furthermore, 5-methylindole significantly potentiates aminoglycoside antibiotics, but not fluoroquinolones, killing of *S. aureus*. In addition, 5-iodoindole also potentiates aminoglycosides. Our findings open a new avenue to develop indole derivatives like 5-methylindole as antibacterial agents or adjuvants of aminoglycoside.

## INTRODUCTION

The discovery and application of antibiotics have dramatically improved the lives of humans and are considered as the greatest medical achievement of last century (*Fleming, 1945*). However, the antibiotic resistance of bacterial pathogens was immediately observed in laboratory beginning in the 1930s (*Fleming, 1945*; *Abraham & Chain, 1940*) and has become a severe threat to global public health and economic development (*Shankar, 2016*; *Lewis, 2010*; *Antimicrobial Resistance Collaborators, 2022*). To counteract this formidable challenge for human society, comprehensive strategies must be adopted. The discovery and development of new antibiotics, including structurally modified existing antibiotics, has played a dominant role in this war (*Brown & Wright, 2016*; *Lewis, 2020*). In addition, improving the efficacy of existing antibiotics has also attracted attentions in the recent decade (*Allison, Brynildsen & Collins, 2011*; *Peng et al., 2015*; *Liu et al., 2019*; *Wright, 2016*; *Zhao et al., 2020*; *Lv et al., 2022a*; *Lv et al., 2022b*), given that these existing antibiotics have

Corresponding authors
Fengqi Sun, 972094096@qq.com
Yajuan Chen, cyj288@fjnu.edu.cn

been well documented in their toxicity, pharmacokinetics, administration and mechanism of actions.

Indole is an intra-species, interspecies or even interkingdom signaling molecule and plays important biological functions in bacteria and animals (*Bansal et al., 2010*; *Lee & Lee, 2010*; *Song & Wood, 2020*). Cumulative evidence suggests that indole is able to reduce the antibiotic tolerance of *Escherichia coli* (*Hu et al., 2015*; *Kwan et al., 2015*) and *Lysobacter enzymogenes* (*Han et al., 2017*; *Wang et al., 2019*) can be dramatically lowered by indole. In addition, indole also reduces *E. coli* biofilm formation (*Domka, Lee & Wood, 2006*; *Domka et al., 2007*; *Lee, Jayaraman & Wood, 2007*). Moreover, indole was reported to increase the bacterial uptake of antimicrobials through an interaction with the Mtr permease (*Zhang, Yang & Defoirdt, 2022*), and indole signaling is a valid target for the development of novel therapeutics in order to control infections caused by Harveyi clade vibrios in aquaculture (*Wu et al., 2022*). These studies suggest that indole may benefit antibiotic treatment when combined with different antibiotics. Furthermore, halogenated indoles were reported to exhibit antibacterial activity against persister cells and biofilms (*Lee et al., 2016*), both of which play critical roles in recurrent infections and antibiotic resistance development (*Lewis, 2007*; *Levin-Reisman et al., 2017*; *de la Fuente-Núñez et al., 2013*; *Liu et al., 2020*; *Boya, Lee & Lee, 2022*). Recently, a substituted indole was found to kill *Pseudomonas aeruginosa* persister cells effectively by damaging membranes and causing lysis (*Song et al., 2019*).

Previously, we found that indole and 5-methylindole could markedly potentiate aminoglycoside antibiotics against bacterial persister cells under hypoionic conditions (*i.e.*, in ion-free solutions) (*Sun et al., 2020*) after short-term combined treatment. Although such potentiation could be fulfilled within 5-min incubation, the requirement of hypoionic conditions for treatment may limit the potential of its application, as salts and electrolytes are ubiquitously present throughout human and animal bodies. In this work, we attempted to overcome this drawback. We show that 5-methylindole could directly kill various Gram-positive and Gram-negative bacteria and also potentiate aminoglycoside antibiotics against stationary-phase *Staphylococcus aureus* cells under conventional treatment conditions (*i.e.*, agitation in LB medium at 37 °C for a few hours). Our study may pave the way to develop indole derivatives as antibacterial agents and/or adjuvants of existing antibiotics.

## MATERIALS AND METHODS

### Strains, medium and reagents

Various Gram-positive (*S. aureus, Staphylococcus epidermidis, Enterococcus faecalis, Streptococcus pyogenes, Micrococcus luteus, Streptococcus iniae*) and Gram-negative (*E. coli, P. aeruginosa* and *Shigella flexneri*) bacterial strains, as well as *M. tuberculosis* strain (H37Ra) were used in this study and their characteristics are described in Table S1. Briefly, each bacterial strain was cultured over-night and then diluted at 1:500 in Luria-Bertani (LB) medium (Note: the MRS medium was used for *E. faecalis*) to further agitated in a shaker (37 °C, 220 rpm) for 18–24 h to prepare stationary-phase cells. All chemical reagents are of analytical purity. Stock solutions of indole, 2-methylindole,

5-methylindole and 5-iodoindole were prepared with DMSO and placed in brown, opaque Eppendorf tubes to avoid photo-induced damage.

## Cell survival assay

Briefly, bacterial cells in stationary-phase were mixed with indole, 2-methylindole, 5-methylindole at varying concentrations and agitated for 3 h. A total of 100 μL treated cell cultures were centrifuged (13,000 rpm, 1 min) in Eppendorf tubes and washed with PBS (0.27 g/L $KH_2PO_4$, 1.42 g/L $Na_2HPO_4$, 8 g/L NaCl, 0.2 g/L KCl, pH 7.4). Afterwards, 5 μL of tenfold serially diluted cell suspension were spot plated onto LB agar dishes for cell survival assay. In addition, stationary-phase and exponential-phase *S. aureus* cell cultures were mixed with aminoglycoside antibiotics in the presence of indole, 2-methylindole or 5-methylindole for varying length of time, and cells were washed with PBS before cell survival assay as described above. Independent experiments were repeated at least three times.

## Minimum inhibitory concentration (MIC) assay

The MICs of 5-methylindole for *S. aureus* ATCC25923, *E. faecalis* ATCC 29212, *S. iniae*, *E. coli* BW25113, *P. aeruginosa* PAO1, *S. flexneri* 24T7T, *K. pneumoniae* KP-D367 were measured by incubating freshly inoculated cultures in LB for 24 h containing varying concentrations of indoles. Results were determined by observing visual turbidity and experiments were performed using at least two independent cultures.

## Preparation and eradication of antibiotic-tolerant *S. aureus* persister cells

Nutrient shift-induced persisters were prepared as previously reported (*Chen et al., 2019*). In brief, over-night cultures of *S. aureus* were diluted at 1:100 into LB medium (37 °C, 220 rpm) and cultured to mid-exponential phase at a cell density of $OD_{600}$ = 0.5–0.6. Cells were centrifuged and washed with M9 medium twice before transferred to M9 medium plus 2 g/L fumarate and agitated for four hours before indole treatment.
Starvation-induced persisters were prepared according to an earlier report (*Eng et al., 1991*). Briefly, *S. aureus* cells were diluted at 1:500 in Mueller-Hinton broth medium and cultured for 24 h (37 °C, 220 rpm) to a cell density of around $10^9$ CFU/mL. Cells were centrifuged, re-suspended in yeast nitrogen broth medium without amino acids, diluted to a cell density of around $10^8$ CFU/mL and agitated for 5 h before indole treatment.

## Hemolysis effect assay

Mouse blood cells were obtained from mice discarded from other animal experiments and sodium citrate at a final concentration of 2.5% was used as anticoagulant. Blood cells were collected by centrifugation (1,000 rpm, 15 min) and washed twice with 0.9% NaCl containing 2.5% sodium citrate. The cell pellets were re-suspended in 0.9% NaCl solution containing 2.5% sodium citrate plus indole, 2-methylindole or 5-methylindole at increasing concentrations, incubated at 37 °C for 3 h and centrifuged, with the supernatant being subjected to absorbance measurement at 545 nm on a 96-well plate reader.

### Ethics statement

Animal experiments were performed in accordance with the National Standards of China (GB/T 35892-2018 and GB/T 35823-2018) and after the approval issued by the Animal Ethical and Welfare Committee of Fujian Normal University (approval no. IACUC 20190006).

## RESULTS

### 5-Methylindole kills various Gram-positive bacteria in stationary-phase

We evaluated the potential bactericidal activity of indole and its two derivatives, *i.e.*, 5-methylindole and 2-methylindole, against typical Gram-positive bacteria, including *S. aureus* and *Staphylococcus epidermidis*. We prepared these bacteria in stationary-phase that are generally tolerant to bactericidal antibiotics (*Keren et al., 2004*) and widely used for antibiotic research (*Allison, Brynildsen & Collins, 2011*; *Peng et al., 2015*), and then subjected them to the treatment with indole derivatives at varying concentrations for 3 h. Cell survival assay revealed that (i) while indole hardly exhibited bactericidal activity, 5-methylindole was able to effectively eradicate these pathogens in a concentration-dependent manner (Figs. 1A–1E); (ii) 2-methylindole eradicated *S. aureus*, *S. epidermidis* and *E. faecalis* as effectively as did 5-methylindole but had no killing effect on *S. pyogenes* and *M. luteus*. We also examined the aquatic pathogen *Streptococcus iniae* and found that 5-methylindole could kill *S. iniae* in a concentration-dependent manner, with 2-methylindole and indole being less effective (Fig. 1F). Furthermore, we determined MICs of 5-methylindole to *S. aureus*, *E. faecalis* and *S. iniae*, which were 4, 16 and 16 mM, respectively (Table 1).

### 5-Methylindole kills various Gram-negative bacteria in stationary-phase

We then examined the potential bactericidal activity of 5-methylindole against typical Gram-negative bacteria in stationary-phase, including *E. coli*, *P. aeruginosa* and *Shigella flexneri*. The cell survival assay showed that 5-methylindole was able to effectively kill all these pathogens in a concentration-dependent manner (Figs. 2A–2C). Similarly, 2-methylindole and indole were also able to kill these pathogens in a concentration-dependent manner, albeit at lower efficacy than 5-methylindole. Notably, Gram-negative bacteria overall seemed more sensitive to 5-methylindole than Gram-positive bacteria (Fig. 2 *vs* Fig. 1). In addition, we determined MICs of 5-methylindole to *E. coli*, *P. aeruginosa* and *Shigella flexneri*, which were 8, 16 and 2 mM, respectively (Table 1).

### 5-Methylindole kills antibiotic-resistant/tolerant bacteria and *Mycobacterium tuberculosis*

Considerable bactericidal activity of 5-methylindole against both Gram-positive and Gram-negative bacterial pathogens, as described in Figs. 1 and 2, prompted us to further examine whether it could eradicate those antibiotic-resistant/tolerant pathogens that give rise to the failure of antibiotic therapy. For this purpose, we first examined

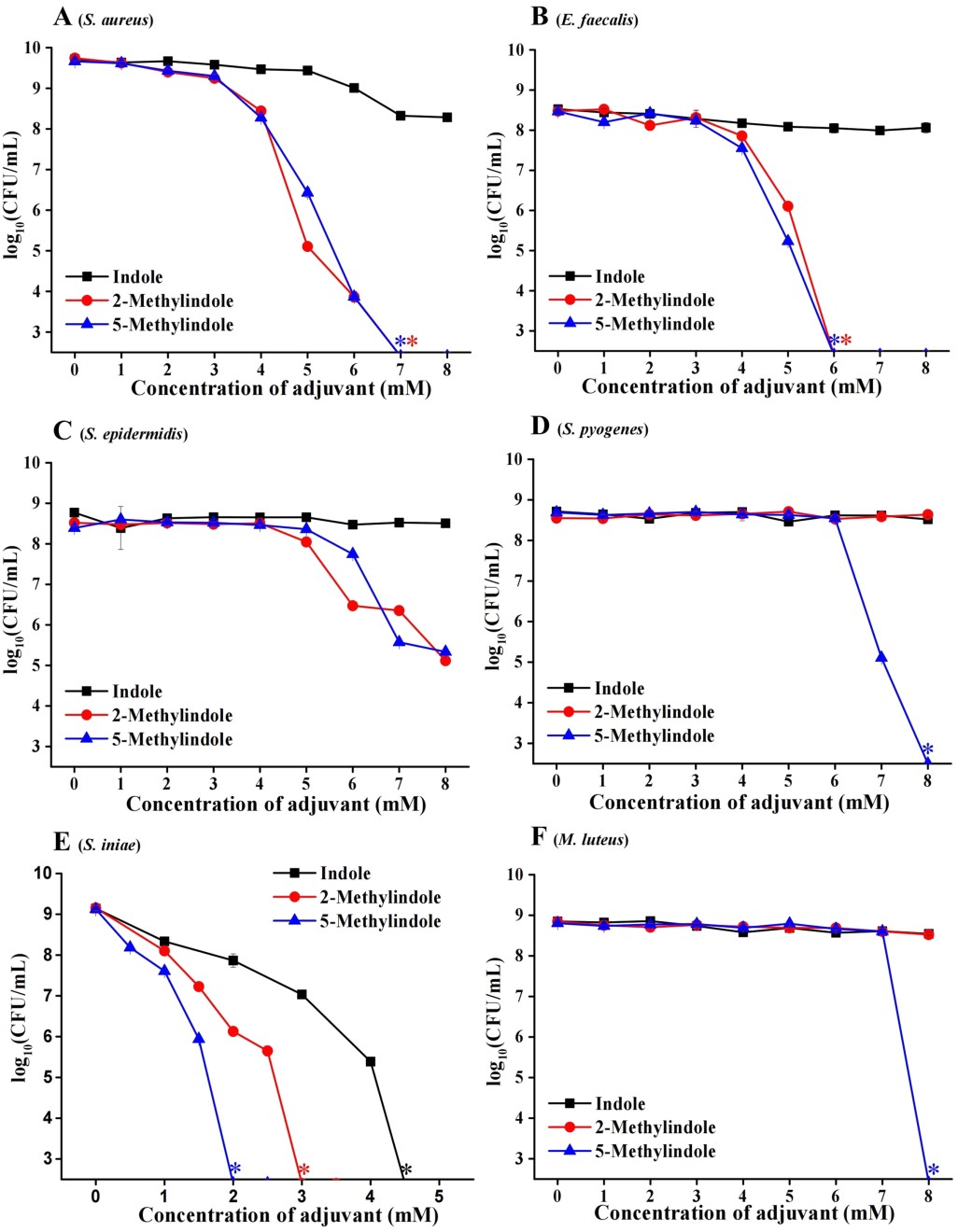

**Figure 1  5-Methylindole exhibited bactericidal activity against Gram-positive bacteria in stationary-phase.** Survival of different Gram-positive bacteria in stationary-phase following a 3-h treatment with indole, 2-methylindole or 5-methylindole at increasing concentrations (0, 1, 2, 3, 4, 5, 6, 7 and 8 mM in Panels (A–D) and F); 0, 0.5, 1, 1.5, 2, 2.5, 3, 3.5, 4 and 4.5 in (Panel E)). Data represent the means ± SD of three replicates of one independent experiment. An asterisk (*) represents no detectable colonies on LB dish when 5 μL treated cells were spot plated.

methicillin-resistant *Staphylococcus aureus* (MRSA) that has been designated on the WHO priority pathogens list (*WHO, 2017*), as well as multidrug-resistant pathogen *Klebsiella pneumoniae* that has been reported as one of the most critical antibiotic-resistant

**Table 1  MICs of 5-methylindole examined in this study.**

| Strain | 5-Methylindole (mM) |
|---|---|
| *S. aureus* ATCC 25923 | 4 |
| *E. faecalis* ATCC 29212 | 16 |
| *S. iniae* | 16 |
| *E. coli* BW25113 | 8 |
| *P. aeruginosa* PAO1 | 16 |
| *S. flexneri* 24T7T | 2 |
| *K. pneumoniae* KP-D367 | 4 |

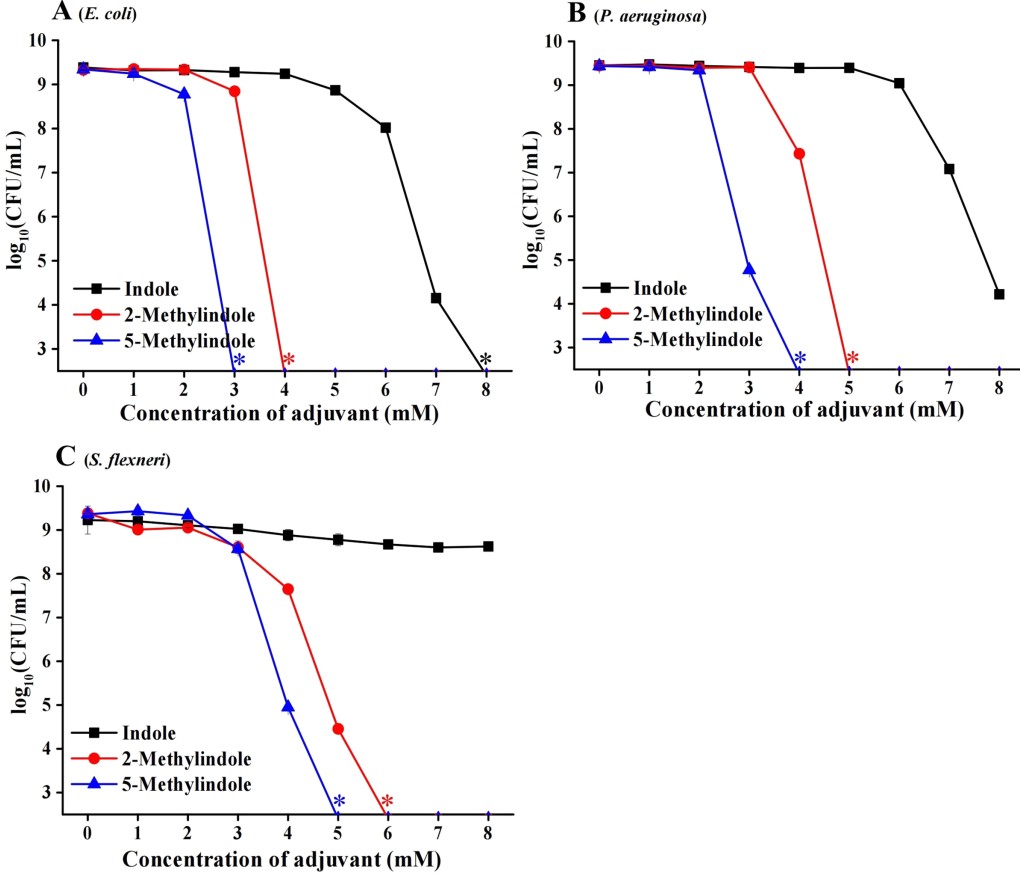

**Figure 2  5-Methylindole exhibited bactericidal activity against Gram-negative bacteria in stationary-phase.** Survival of different Gram-negative bacteria in stationary-phase following a 3-h treatment with indole, 2-methylindole or 5-methylindole at increasing concentrations (0, 1, 2, 3, 4, 5, 6, 7 and 8 mM). Data represent the means ± SD of three replicates of one independent experiment. An asterisk (*) represents no detectable colonies on LB dish when 5 μL treated cells were spot plated.

pathogens in China (*China Antimicrobial Resistance Surveillance System, 2020*). The cell survival assay showed that 5-methylindole was able to eradicate these two antibiotic-resistant pathogens in a concentration-dependent manner (Figs. 3A and 3B)

while 2-methylindole was less effective. Notably, we found that 5-methylindole and 2-methylindole were also able to kill *Mycobacterium tuberculosis* H37Ra (an avirulent strain) in a concentration-dependent manner (Fig. 3C). In addition, MIC of 5-methylindole to *K. pneumoniae* was determined to be 4 mM (Table 1).

We next examined the effect of 5-methylindole on bacterial persisters, which are transiently tolerant to the attack of bactericidal antibiotics and play an important role in chronic infections and the evolution of antibiotic resistance (*Lewis, 2010*; *Levin-Reisman et al., 2017*; *Liu et al., 2020*). To this end, nutrient shift-induced *S. aureus* persisters were prepared by switching the carbon source of *S. aureus* cells in exponential-phase to fumarate (*Radzikowski et al., 2016*), and starvation-induced *S. aureus* persisters were made by transferring *S. aureus* cells in stationary-phase to medium without any nutrients (*Chen et al., 2019*). These nutrient shift-induced and starvation-induced *S. aureus* persister cells were kept in non-replicating status and highly tolerant to the attack by tobramycin under conventional treatment condition (Fig. S1; or refer to our earlier reports (*Sun et al., 2020*; *Chen et al., 2019*)). However, both of them were killed by 5-methylindole and 2-methylindole in a concentration-dependent manner, with indole being much less effective (Figs. 3D and 3E).

## 5-Methylindole potentiates aminoglycosides killing of conventional *S. aureus* and MRSA cells in stationary-phase but not the cells in exponential-phase

We previously reported that 5-methylindole could potentiate aminoglycoside antibiotics against conventional *S. aureus* and MRSA cells under hypoionic conditions (*i.e.*, in ion-free solutions) and that such potentiation could be achieved within a couple of minutes (*Sun et al., 2020*). Here we examined whether 5-methylindole could potentiate aminoglycosides under conventional treatment conditions (*i.e.*, agitation in LB medium for a couple of hours). The cell survival assay showed that 5-methylindole could significantly enhance the bactericidal actions of tobramycin against conventional *S. aureus* cells in stationary-phase, in both time-dependent (Fig. 4A) and concentration-dependent manners (Fig. 4B). In addition, tobramycin alone at 100 µg/mL only exhibited marginal bactericidal activity, but eradicated *S. aureus* cells by more than four orders of magnitude when combined with 5-methylindole (Fig. 4C). More importantly, we found that both 5-methylindole and 2-methylindole significantly increased the killing efficacy of streptomycin against MRSA cells, with streptomycin alone hardly killing the bacteria (Fig. 4D). We also observed that 5-methylindole, 2-methylindole and indole could dramatically potentiate gentamicin, streptomycin and kanamycin against *S. aureus* cells in stationary-phase (Figs. S2B–S2D). However, they had little effects on fluoroquinolone antibiotics ofloxacin and ciprofloxacin (Figs. S2E and S2F), in contrast to earlier reports showing that ofloxacin was potentiated by ascorbic acid against *Staphylococcus aureus* (*Dey & Bishayi, 2018*) and certain oils could reduce the MIC of ofloxacin against *E. coli* and *S. aureus* (*Pereira et al., 2020*). In addition, we found that none of 5-methylindole, 2-methylindole and indole could potentiate tobramycin and other types of aminoglycosides against *S. aureus* cells in exponential-phase (Figs. S3). Together, these results suggest that

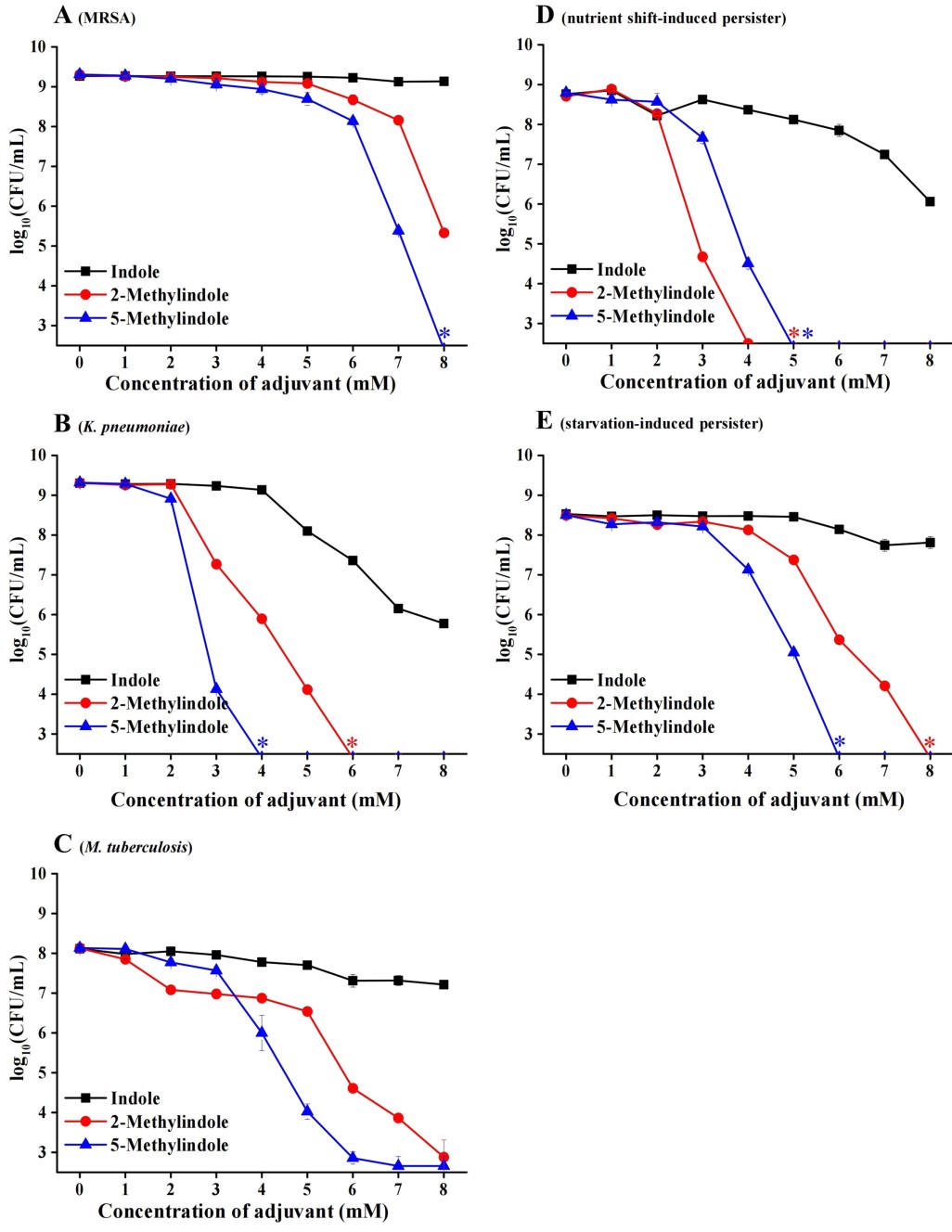

**Figure 3** **5-Methylindole exhibited bactericidal activity against antibiotic-resistant/tolerant bacteria.**
(A–C) Survival of MRSA (panel A), *K. pneumoniae* (panel B) and *M. tuberculosis H37Ra* (an avirulent strain, panel C) in stationary-phase following a 3-h treatment with indole, 2-methylindole or 5-methylindole at increasing concentrations (0, 1, 2, 3, 4, 5, 6, 7 and 8 mM). (D, E) Survival of *S. aureus* nutrient shift-induced (panel D) and starvation-induced persister cells (panel E) following a 3-h treatment with indole, 2-methylindole or 5-methylindole at increasing concentrations (0, 1, 2, 3, 4, 5, 6, 7 and 8 mM). Data represent the means ± SD of three replicates of one independent experiment. An asterisk (*) represents no detectable colonies on LB dish when 5 μL treated cells were spot plated.

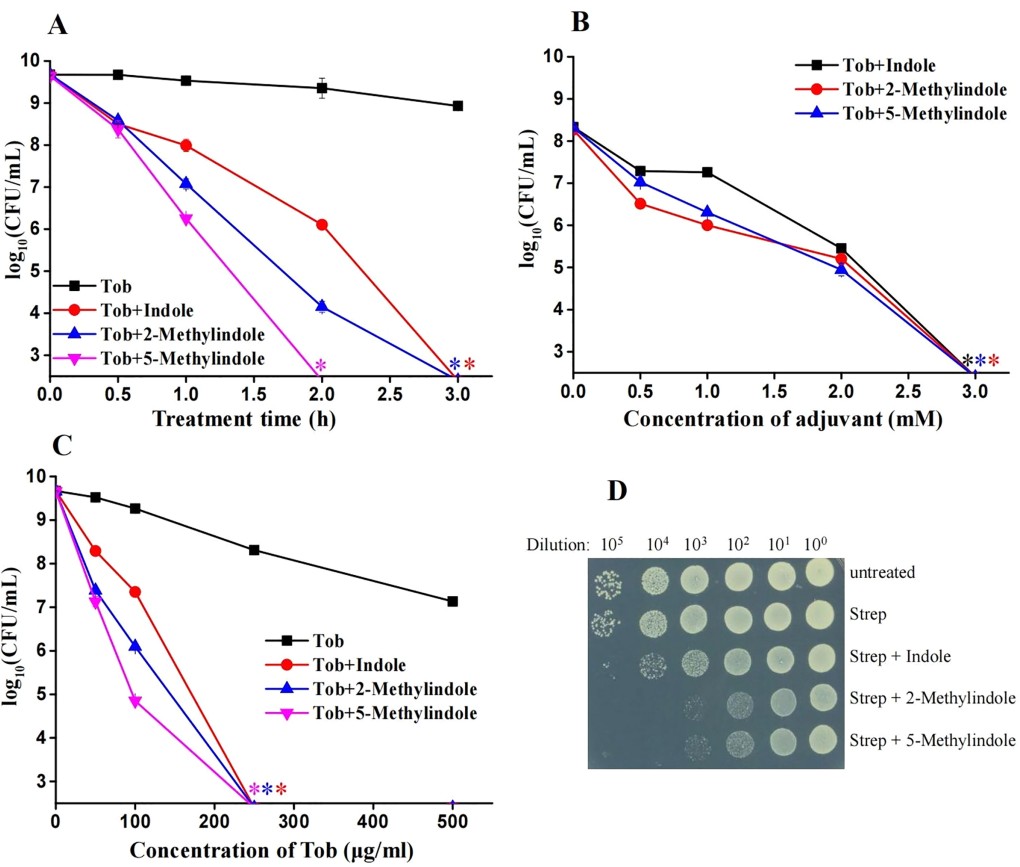

**Figure 4** **5-Methylindole potentiates aminoglycosides to kill conventional *S. aureus* and MRSA cells in stationary-phase.** (A) Survival of *S. aureus* cells in stationary-phase following a combined treatment with 250 µg/mL tobramycin plus 4 mM indole, 2-methylindole or 5-methylindole for varying length of time (0.5, 1, 2 and 3 h). (B) Survival of *S. aureus* cells in stationary-phase following a 3-h combined treatment with 250 µg/mL tobramycin plus indole, 2-methylindole or 5-methylindole at increasing concentrations. (C) Survival of *S. aureus* cells in stationary-phase following a 3-h combined treatment with increasing concentrations of tobramycin (0, 50, 100, 250 and 500 µg/mL) plus 4 mM indole, 2-methylindole or 5-methylindole. (D) Survival of MRSA cells in stationary-phase following a 3-h combined treatment with 500 µg/ml streptomycin plus 4 mM indole, 2-methylindole or 5-methylindole. Data in panels A, B and C represent the means ± SD of three replicates of one independent experiment and an asterisk (*) represents no detectable colonies on LB dish when 5 µL treated cells were spot plated.

5-methylindole could potentiate aminoglycoside antibiotics against *S. aureus* cells in stationary-phase.

## 5-Methylindole at sub-toxic concentrations potentiates aminoglycosides killing of *S. aureus* in stationary-phase

Because 5-methylindole alone could kill bacteria, we thus evaluated its cytotoxicity towards mammalian cells by examining its hemolysis effect. Cell lysis assay revealed that 5-methylindole could disrupt mouse blood cells in a concentration-dependent manner (Fig. 5A, Fig. S4), with the hemolysis effect of 1 mM 5-methylindole being minimal. In comparison, the hemolysis effects of 2-methylindole and indole were less potent than

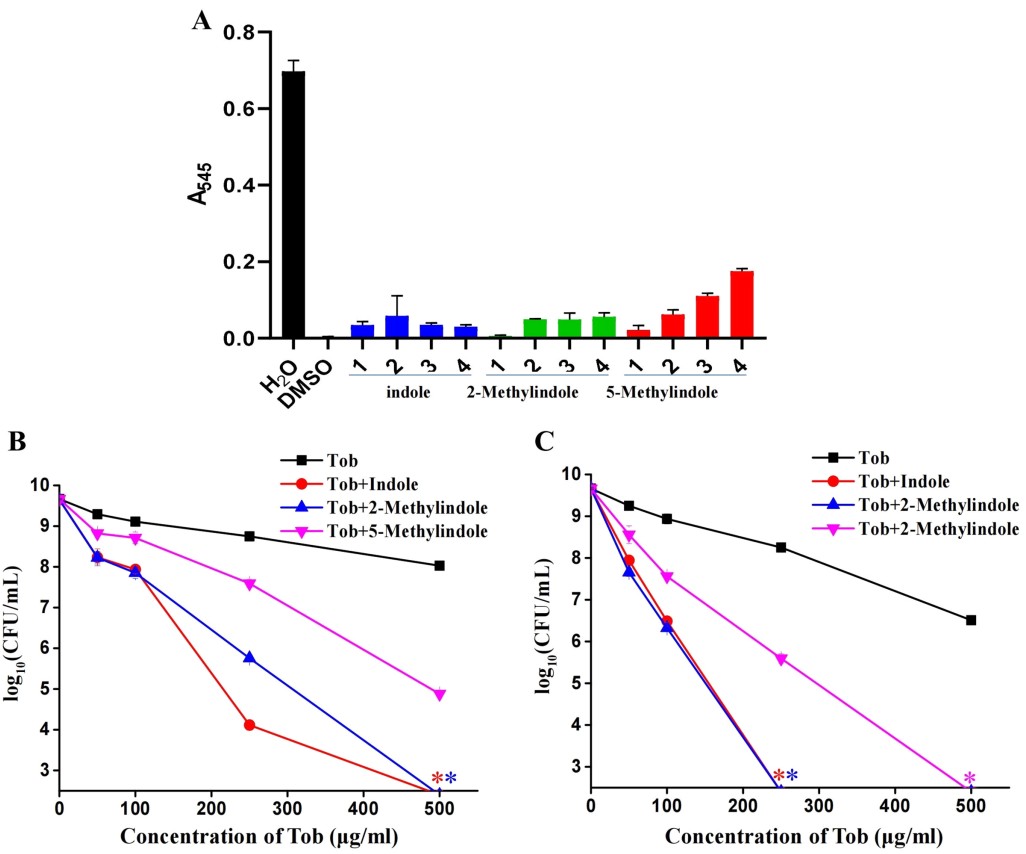

**Figure 5  5-Methylindole at sub-toxic concentration potentiates tobramycin killing of *S. aureus* in stationary-phase.** (A) Absorbance at 545 nm of the supernatant of mouse blood cell suspension following a 3-h treatment with indole, 2-methylindole or 5-methylindole at increasing concentrations. (B, C) Survival of *S. aureus* cells in stationary-phase following a 3-h (panel B) or 5-h (panel C) combined treatment with increasing concentrations of tobramycin (0, 50, 100, 250 and 500 μg/mL) plus 4 mM indole, 2 mM 2-methylindole or 1 mM 5-methylindole. Data represent the means ± SD of three replicates of one independent experiment. An asterisk (*) represents no detectable colonies on LB dish when 5 μL treated cells were spot plated.                 

that of 5-methylindole. Based on this, we then evaluated the potentiation effects of 5-methylindole, 2-methylindole and indole at sub-toxic concentrations (1, 2 and 4 mM, respectively) on tobramycin. The cell survival assay revealed that all of them could effectively enhance the killing efficacy of tobramycin against *S. aureus* cells in stationary-phase after agitation for 3 h (Fig. 5B) or 5 h (Fig. 5C).

## DISCUSSION

In comparison with our earlier report (*Sun et al., 2020*), several novel findings have been made in the current work. First, we observed that 5-methylindole itself exhibited substantial antibacterial activities against a number of Gram-positive and Gram-negative bacterial pathogens, including antibiotic-tolerant *S. aureus* persisters, MRSA and *M. tuberculosis*. Second, we observed the potentiation effect of 5-methylindole on aminoglycoside antibiotics against *S. aureus* cells under conventional treatment conditions (*i.e.*, agitation in LB medium for several hours). In contrast, the potentiation effect of 5-

methylindole on aminoglycosides was detected only under hypoionic conditions (*i.e.*, without the presence of ions and electrolytes) in our earlier report (*Sun et al., 2020*), apparently being distinct from physiological conditions. In addition, this unique treatment lasts only a few minutes whereas the treatment in the current study lasts several hours, which is more similar to antibiotic chemotherapy. Therefore, the current work further illustrates the potential of 5-methylindole for the development of new antibiotics and/or antibiotic adjuvants.

Indole exerts diverse roles in multiple signaling pathways of bacteria as an endogenous metabolite, such as spore formation, plasmid stability, drug resistance, biofilm formation, and virulence in indole-producing bacteria (*Lee & Lee, 2010*). In a rich medium (*Lee, Jayaraman & Wood, 2007*), *E. coli* can generate endogenous indole up to 0.6 mM, which exhibits no toxicity to *E. coli* (Fig. 2A) but can induce persister formation (*Vega et al., 2012*). Moreover, indole can mediate the signaling cross-talk between enteric bacteria and their host (*Bansal et al., 2010*). Notably, indole and some of indole-based derivatives have been reported to exhibit antibacterial, antibiofilm and/or antivirulence activities (*Song & Wood, 2020*; *Lee et al., 2016*; *Chimerel et al., 2012*; *Qin et al., 2020*; *Sethupathy et al., 2020*). In retrospect, 2-methylindoe, but not 5-methylindole, was found to inhibit biofilm formation of an opportunistic fungal pathogen *Candida albicans* (*Lee et al., 2018*); 7-methylindole was reported to significantly suppress biofilm formation of the Gram-negative pathogen *Serratia marcescens* (*Sethupathy et al., 2020*). In particular, 5-iodoindole and 5-chloroindole were reported to exhibit antibiofilm and antibacterial activities (*Lee et al., 2016*; *Boya, Lee & Lee, 2022*). Prompted by these reports, we examined 5-iodoindole and found that this indole derivative could also potentiate tobramycin killing of stationary-phase *S. aureus* cells (Fig. S5A). Notably, the potentiation effect of 5-iodoindole on tobramycin appears to be higher than that of 5-methylindole, presumably due to its stronger antibacterial effect than the latter (Fig. S5B). Together, these observations illustrate a high potential of methylindole derivatives in the discovery and development of new antibiotics.

Besides antibacterial activity, indole and its derivatives have also been shown to be able to change the efficacy of existing antibiotics in killing of bacterial pathogens. For instance, indole has been reported to increase the antibiotic tolerance of *E. coli* cells (*Lee et al., 2010*; *Vega et al., 2013*; *Han et al., 2011*). Such suppression effect might be achieved by elevating the expression of efflux pumps that could transport the intracellular antibiotics out of the bacterial cells (*Hirakawa et al., 2005*) and/or activating stress response-related pathways that could counteract the lethality of the antibiotics (*Vega et al., 2012*). Nevertheless, several studies revealed that indole can reduce the antibiotic tolerance of *E. coli* (*Hu et al., 2015*; *Kwan et al., 2015*) and *Lysobacter enzymogenes* (*Han et al., 2017*; *Wang et al., 2019*), likely by activating or upregulating antibiotic importers. These contradictory observations might result from the difference in cell growth/metabolic status, as we observed that 5-methylindole potentiates aminoglycoside antibiotics under hypoionic conditions against *S. aureus* cells in stationary-phase but suppress the antibiotics against the cells in exponential-phase (*Sun et al., 2020*). In support of this, here we show that 5-methylindole

potentiates tobramycin against *S. aureus* cells in stationary-phase but had no effects on the antibiotics against the cells in exponential-phase.

The molecular mechanism by which indole and 5-methylindole kills bacterial pathogens and potentiates aminoglycosides remains unknown. One possibility is that indole and 5-methylindole may reduce the electrochemical potential across the cytoplasmic membrane of bacterial cells, as indole was reported to act as a proton ionophore and thereby prevent formation of the FtsZ ring that is a prerequisite for cell division (*Chimerel et al., 2012*). Another possibility is that 5-methylindole may activate certain membrane channels for enhancing aminoglycoside uptake, as indole was reported to increase the antibiotic susceptibility of *Lysobacter enzymogenes* (*Lewis, 2020*; *Wang et al., 2019*) by up-regulating antibiotic importers. Aminoglycosides have been widely applied to cure various infections in humans and animals (*van Duijkeren et al., 2019*) and their applications in clinics are largely limited due to the toxicity and the wide spread of aminoglycoside resistance (*Mingeot-Leclercq & Tulkens, 1999*; *Mingeot-Leclercq, Glupczynski & Tulkens, 1999*). In this regard, 5-methylindole-induced aminoglycoside potentiation against conventional *S. aureus* and MRSA cells, under both hypoionic conditions as reported earlier by us (*Sun et al., 2020*) and conventional treatment conditions as seen here, is of clinical importance. Dissection of the underlying mechanism may help to develop indole derivatives as aminoglycoside adjuvants, which would extend the lifetime of this important class of bactericidal antibiotics.

## ACKNOWLEDGEMENTS

We thank Profs. Luhua Lai, Xiaoyun Liu, Zengyi Chang and Dr. Xiaoyun Liu (all from Peking University), Dr. Qingeng Huang (Fujian Normal University) and Prof. Xuanxian Peng (SUN YAT-SEN University) for their kindness in providing bacterial strains as described in Table S1.

### Funding

This work was supported by research grants from the Natural Science Foundation of Fujian Province (2019J01278 to Yajuan Chen), the National Natural Science Foundation of China (No. 31972918 to Xinmiao Fu) and the Natural Science Foundation of Fujian Province (No. 2021J02029 to Xinmiao Fu). The funders had no role in study design, data collection and analysis, decision to publish, or preparation of the manuscript.

### Grant Disclosures

The following grant information was disclosed by the authors:
Natural Science Foundation of Fujian Province: 2019J01278.
National Natural Science Foundation of China: 31972918.
Natural Science Foundation of Fujian Province: 2021J02029.

## Competing Interests

The authors declare that they have no competing interests.

## Author Contributions

- Zhongyan Li performed the experiments, analyzed the data, prepared figures and/or tables, and approved the final draft.
- Fengqi Sun conceived and designed the experiments, performed the experiments, analyzed the data, prepared figures and/or tables, and approved the final draft.
- Xinmiao Fu conceived and designed the experiments, authored or reviewed drafts of the article, and approved the final draft.
- Yajuan Chen conceived and designed the experiments, analyzed the data, authored or reviewed drafts of the article, and approved the final draft.

## Animal Ethics

The following information was supplied relating to ethical approvals (*i.e.*, approving body and any reference numbers):

This study was approved by the Animal Ethical and Welfare Committee of Fujian Normal University (IACUC 20190006).

## Data Availability

The raw measurements are available in the Supplemental Files.

## Supplemental Information

Supplemental information for this article can be found online at http://dx.doi.org/10.7717/peerj.14010#supplemental-information.

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
