# Peer review of "-Methylindole kills various bacterial pathogens and potentiates aminoglycoside against methicillin-resistant Staphylococcus aureus"

_PeerJ, doi:10.7717/peerj.14010_

## Round 0.1 · original submission · Major Revisions

Dear authors, in agreement with both reviewers, I think your manuscript needs major changes before it can be accepted for publication.

Reviewer 1 ·

Basic reporting

Current work is interesting due to provides information about potencial molecules with antimicrobial activity.
Author makes emphasis on derivates of indole which is known has antimicrobial activity.
However, has several methodological details.
In abstract, after Particulary, 5-Methhlindole must be changed by 5-methylindole.
line 25, same error.
line 38 that is double.

Experimental design

in the experimental design there are several issues.
First to all. Why do author prefers to do dilution instead up a well controlled experiment were they represent microbial charge with OD in order to have a strict and rigorous experiment for all microorganisms tested, same charge, same conditions, how do they can be absolutely sure that all microorganisms had the same final Over night charge? This must be perfectly controlled.
Line 75: Cell survival assay, author must give in this part what were all concentrations tested, not only in figures because this is a medular part of the study.
Which concentration of amynoglcosides were tested? why experiments were done in a kind of drop onto the plate? The best way to prove activity in a quatintative manner is with the broth microdilution method due we dan do minimal inhibitory concentrations Why they have not control on inoculum?
Line 91: How author are sure that dilution 1:500 is approximately 10exp9 CFU/mL? How are they can be sure that all microorganisms tested grew up at the same kynetic?
Line 170: The use of 100 ug/mL is huge concentration, how do author decided to use thatr concentration? Please to check EUCAST and CLSI M100 in order to see what concentration is defined and correlationated with clinical values.

Validity of the findings

Current work is interesitng however experiments control (I meant, strict and rigorous) is needed.
A raw data in excel is shared however it is not mentioned in any part of the paper where the date cames from. Results are not representative due there is not control in the grew of microorganisms beside growth curve are not showed.
With the rigor of experiments a lot of bias could be present

Annotated reviews are not available for download in order to protect the identity of reviewers who chose to remain anonymous.

·

Basic reporting

Authors report the antibacterial activity of 5-methylindole against several bacteria including Staphylococcus aureus. Overall, the manuscript is interested, but not thoroughly studied and discussed. Here are several issues to improve.

Experimental design

There is a novelty issue since a very similar study was published by the same group (5-Methylindole Potentiates Aminoglycoside Against Gram-Positive Bacteria Including Staphylococcus aureus Persisters Under Hypoionic Conditions, Front Cell Infect Microbiol, 2020: 10: 84). The difference between and this work and the previous work looks different assay conditions used. And two results are similar and even many sentences are same. Please discuss clearly what advances were achieved from this study and check copy killer carefully to remove same sentences including all methods.

Validity of the findings

Authors investigated indole, 2-methyindole, and 5-methylindole at high concentrations. This reviewer wonders why authors selected only three compounds since it appears that 5-iodoindole was better in the previous study (DOI 10.1186/s13568-016-0297-6).
Another concern is that doses used are pretty high as 4 and 7 mM of 2-methyindole, and 5-methylindole are active. Even 1-4 mM of 5-methylindole could disrupt mouse blood cells in a concentration-dependent manner (Figure 5). Hence the cytotoxicity problem appears severe. Recent new antibiotics requires MICs less than 0.1 mg/mL and current MICs are much higher with chemical cytotoxicity. Some cytotoxicity assay with at least animal cells is required. Also, mention its active concentrations in the abstract.
Authors insisted that 5-methylindole specifically potentiates aminoglycoside antibiotics, but not fluoroquinolones. It is not clear how authors concluded this since fluoroquinolones alone did not kill S. aureus so that it is difficult for 5-methylindole to enhance. Better to remove “specifically”. Please discuss why and how 5-methylindole potentiates aminoglycoside antibiotics.
The molecular mechanism is unknown and there is not much insight on the mechanism. Wondered why 5-methylindole is more active than 2-methylindole and indole.

Additional comments

Please add most recent (2020-2022) references of indole derivatives against various microbes and emphasize why 5-methylindole has specific merit.

---

## Round 0.2 · Minor Revisions

Dear authors, please make the minor modifications requested by one of the reviewers in order to make it suitable for publication, thanks.

Reviewer 1 ·

Basic reporting

Author have made several changes according previous recommendations.
Line 47. To change E. coli by Escherichia coli due is the first time author mentioned the microorganism.
Line75. Title says "Bacterial pathogens" however whithin bacteria choosen such experimental model is included Micrococcus luteus which participation such pathogen the most of time is debatible or questionable, on the other hand same to Streptococcus iniae whith very few reports in humans.
Line 76. To change Pseudomonas aeruginosa to P. aeruginosa due is the second time mentioned.
Line 76. To specify M. tuberculosis strain (H37Ra).
Line 99, bacteria names must be in italic.
In section Section Hemolysis effect assay, I´d like to know why author decided to add anticoagulant and what anticoagulant did they use. Blood coagulation is due a several factors that are in plasma, however, when a wash is made all these factors are removed, beside the previous efect of anticoagulant itself, it is know the effect that certain anticoagulants have on microorganisms.
Line 211. To change ciproxacin by ciprofloxacin.
Line 263. After Fig. s5B and space between dot and Together.
Could you include how Escherichia coli could to respond the indol effect due this microorganism is indole producer naturally when tryptophan is present. Add in discussion something about this, please.
Somew images are in chinese characters, could you change it please.

Experimental design

Author have made several changes according previous recommendations and added new ones.
For the section named minimum inhibitory concentrations. Interpretation is made visually with the use of any equipment such a spectophotometer, this according several guides such CLSI and EUCAST, thus, remove measuring optical density (OD) at 600 nm.
Section Preparation and eradication of antibiotic-tolerant S. aureus persister cells
To specify concentration of tobramycin used.

Validity of the findings

Several changes have been made with the methodological rigor.

·

Basic reporting

The revised manuscript has been improved.

Experimental design

The revised manuscript has been improved.

Validity of the findings

The revised manuscript has been improved.

---

## Round 0.3 · accepted · Accept

Thanks for addressing the reviewer´s comments.